# Tackling Pharmaceutical Pollution Along the Product Lifecycle: Roles and Responsibilities for Producers, Regulators and Prescribers

**DOI:** 10.3390/pharmacy12060173

**Published:** 2024-11-22

**Authors:** Gillian Parker, Fiona A. Miller

**Affiliations:** 1Institute of Health Policy, Management and Evaluation, Dalla Lana School of Public Health, University of Toronto, Toronto, ON M5T 3M6, Canada; gillian.parker@utoronto.ca; 2Collaborative Centre for Climate, Health & Sustainable Care, University of Toronto, Toronto, ON M5T 3M6, Canada

**Keywords:** pharmaceuticals, environmental, lifecycle, pollution, healthcare, producers, regulators, prescribers, climate change

## Abstract

Pharmaceuticals produce considerable environmental harm. The industry’s resource-intensive nature, coupled with high energy costs for manufacturing and transportation, contribute to the “upstream” harms from greenhouse gas emissions and ecosystem pollution, while factors such as overprescription, overuse, and pharmaceutical waste contribute to the “downstream” harms. Effectively addressing pharmaceutical pollution requires an understanding of the key roles and responsibilities along the product lifecycle. In this commentary, we argue that three actors—producers, regulators, and prescribers—have unique and interdependent responsibilities to address these issues. Producers and market access regulators are upstream actors who can manage and mitigate harms by both shifting manufacturing, business practices, and regulatory requirements and producing transparent, robust data on environmental harms. By contrast, prescribers are downstream actors whose capacity to reduce environmental harms arises principally as a “co-benefit” of reducing inappropriate prescribing and overuse. Potentially complicating the prescriber’s role are the calls for prescribers to recommend “environmentally preferable medicines”. These calls continue to increase, even with the sparsity of transparent and robust data on the impact of pharmaceuticals on the environment. Recognizing the interdependencies among actors, we argue that, rather than being ineffectual, these calls draw needed attention to the critical responsibility for upstream actors to prioritize data production, reporting standards and public transparency to facilitate future downstream efforts to tackle pharmaceutical pollution.

## 1. Introduction

Already high, the use of pharmaceuticals is expected to continue to rise due to factors such as aging populations, increased life expectancy, economic growth, intensified livestock and aquaculture practices, and the exacerbation of existing diseases by climate change [1]. The entire pharmaceutical lifecycle, from production to use and disposal, is responsible for myriad environmental harms: most significantly, greenhouse gas (GHG) emissions and ecotoxicity. The global pharmaceutical industry is a major contributor to GHG emissions [2]. Indeed, pharmaceuticals, at approximately 20–25%, are estimated to account for the largest single component of healthcare’s total emissions [3,4,5]. In addition to contributing to climate change, pharmaceutical consumption, through prescription medicines, over-the-counter therapeutic drugs, and veterinary drugs, release thousands of active ingredients into waterways, polluting ecosystems worldwide [1,6,7,8,9]. With clear environmental harms from pharmaceuticals, the responsibilities for addressing them are complex and shared among actors along the product lifecycle.

In this commentary, we argue that three actors—producers, regulators, and prescribers—have differently weighted responsibilities to address pharmaceutical pollution. These responsibilities include resolving the lack of rigorous and transparent data and addressing emerging efforts to select environmentally preferable medicines. In what follows, we elaborate this argument. First, we describe the significant pharmaceutical contributions to GHG emissions and ecotoxicity pollution. Then, we discuss actions for upstream actors to manage, mitigate, and report on environmental harms, followed by how the most impactful downstream actions to reduce pharmaceutical pollution should focus on appropriately prescribing medicines.

## 2. Understanding Pharmaceutical Pollution: GHG Emissions and Ecotoxicity

To fully understand how these key actors can address depollution, it is essential to examine two primary environmental impacts from pharmaceuticals: GHG emissions, driven largely by production processes; and ecotoxicity, a result of pharmaceutical waste entering ecosystems.

### 2.1. Pharmaceutical Contributions to Climate Change

Pharmaceuticals are among the most carbon-intensive components of healthcare [3]. The global biotechnology and pharmaceutical industry’s carbon footprint is reported to be greater than many other major industries, including forestry, paper, and semiconductors [10]. Emissions from some medical and anesthetic gases, such as the hydrofluorocarbons (HFCs) in metered dose inhalers, or the volatile anesthetics (e.g., nitrous oxide (N_2_O) and halogenated hydrocarbons) used in surgery, are well known [11,12,13]. Also of note, nearly 99% of pharmaceutical feedstocks (raw material that is required for industrial processes) and reagents are derived from petrochemicals, which contributes to climate change [14]. For example, the synthesis of drugs like aspirin and penicillin relies on organic molecules sourced from petroleum, and polymers derived from petrochemicals are integral to pill capsule and coating production [15]. In addition to their contribution to GHG emissions, pharmaceuticals produce significant amounts of ecotoxic pollution.

### 2.2. Pharmaceutical Pollution in the Ecosystem

Concerns about the ecotoxicity of pharmaceuticals in the environment are longstanding. Because active pharmaceutical ingredients (APIs)—the biologically active ingredient of a medicine that produces the intended effects—are designed to be stable and bioavailable, they persist and accumulate in ecosystems, harming many species and creating risk to human health through exposure via drinking water and the consumption of contaminated food sources [1,16,17,18,19]. While pharmaceuticals are released into the environment as a result of manufacturing accidents or improper disposal, the use-phase release of pharmaceuticals into the ecosystem is, far and away, the most significant contribution to pharmaceutical pollution [20]. Active ingredients in pharmaceuticals that are consumed are excreted via urine and feces, with a sizable portion of these doses—ranging from 30 to 90%—excreted as still-active substances [1,21]. Moreover, as the vast majority of wastewater and drinking water treatment facilities do not remove pharmaceutical ingredients [21,22], this results in high concentrations of pharmaceutical ingredients in water ways downstream from wastewater treatment facilities [23,24]. In addition, the emission of antimicrobials (through veterinary and human use) into the environment is a significant issue as it contributes to the development of antibiotic-resistant bacterial strains, exacerbating antimicrobial resistance, which is a global threat to public health and economic development [25,26,27]. These considerable environmental impacts occur along the product lifecycle and require action from multiple actors to address these harms.

## 3. Key Actors: Producers, Regulators, and Prescribers

Tackling pharmaceutical pollution requires an understanding of which actors can influence which components of the pharmaceutical system to drive depollution. GHG emissions from pharmaceuticals are produced at all stages of the lifecycle, and the emissions “hotspots” vary greatly depending on the product, administration methods, producer, and jurisdiction of production. Similarly, the use-phase contribution to ecotoxicity through the excretion of consumed pharmaceutical products relies on upstream actions such as drug production, regulatory approval, healthcare system purchase, and prescription [21,27]. Three key actors—producers, regulators, and prescribers—within the pharmaceutical system have significant opportunity to reduce pollution from pharmaceuticals (Figure 1).

### 3.1. Producers

To date, the major focus for producers has been to reduce the energy intensity of the production process, but depollution activities need also to prioritize developing “greener” drugs, more sustainable industry operations, and increasing data production and transparency.

#### 3.1.1. Develop “Greener” Pharmaceuticals

The development of environmentally sustainable, greener pharmaceuticals has been suggested as a key strategy to reduce the downstream harm from pharmaceuticals in the environment [28,29]. Experts suggest strategies such as substituting problematic APIs and excipients (the inactive substance that serves as the medium for a drug in the pharmaceutical product) with greener substances, developing pharmaceutical products with enhanced absorption capabilities and improving drug delivery methods to enable smaller, more efficient doses [21,28,29,30]. The concept of greener pharmaceuticals includes production and manufacturing activities that also reduce GHG emissions. Green chemistry techniques, like using renewable raw materials or running reactions at room temperature, can minimize chemical waste and emissions in the manufacturing processes [31]. Additionally, enhancing the temperature stability of pharmaceutical products can lead to reduced energy consumption during storage and transportation [11].

#### 3.1.2. More Sustainable Industry Operations

In addition to shifts to greener medicines, producers need to turn their attention to the environmental impacts of the structural aspects of the pharmaceutical and biotechnology industries’ operations [32]. In particular, the environmental impacts of commercial models need to be understood and addressed; these include large global salesforces and aggressive marketing practices that encourage over-use; and financialized approaches to product development, including via late-stage acquisition, “me-too” innovation, and product extensions. Responsible industry innovation to reduce the environmental impact of products could add significant clinical value, though the example provided by inhalers in the early 2000s, as industry moved away from chlorofluorocarbons (CFCs) with limited R&D investments alongside high prices, suggests the need for substantial regulatory reform [33]. Sustainable industry operations must be pursued and monitored closely to ensure that the changes meaningfully contribute to upstream depollution efforts.

#### 3.1.3. Increase Data Production and Transparency

The production, availability, and transparency of robust environmental data on pharmaceutical ingredients, products, and processes needs to occur concurrently with these producer initiatives to reduce the environmental impact of pharmaceuticals. For example, there is a significant lack of research, transparency, and consistency in the reporting of GHG emissions data for the vast majority of pharmaceuticals [34,35]. While some drug companies have begun to produce data on product specific GHG emissions (typically through life cycle assessments, a type of environmental impact assessment that spans the product or process lifecycle), these assessments are rarely publicly available due to the proprietary nature of the production process [36]. Of note, the emergence of industry-run data portals offer promise, but they also raise questions around the ability of industry to address these critical needs for robust, consistent, and unbiased evidence. Without transparency about pharmaceutical ingredients, products, and processes, comparative reviews of the environmental impact of pharmaceuticals cannot occur.

### 3.2. Regulators

Legislation and polices are critical upstream mechanisms to manage and mitigate the environmental harms from pharmaceuticals, and regulators can increase their impact through enhancing environmental risk assessment (ERA) processes and incorporating GHG emissions reporting into market access requirements. In addition, similar to producers, regulators have been slow to prioritize data production and availability.

#### 3.2.1. Improve Environmental Risk Assessment Processes

Market access regulators, such as the US Food & Drugs Administration, the European Medicines Agency, and Health Canada’s Health Products and Food Branch, have a critical role in setting the terms under which pharmaceuticals are made available for sale. Market access processes for new pharmaceutical substances or products (both prescribed and OTC) in the US, EU, and Canada, currently require environmental risk assessments (ERA). Though critically important, to date, regulatory efforts to manage and mitigate ecotoxicological harms from pharmaceuticals face major limitations, which a substantial body of literature has documented [9,37,38]. Revisions are planned (Proposed Amendments to EU MA legislation) or underway (Canadian Bill S5), including completing ERAs for legacy, on market products and incorporating antimicrobial resistance (AMR)-specific criteria (in proposed EU legislation). While essential, these modernization efforts need to go further to make a meaningful impact on pharmaceutical pollution. For example, requiring remediation based on ERA results, incorporating ERA into post-access processes [9,17,39], and leveraging good manufacturing practices (GMP) [37] hold particular promise. Currently, legislated ERAs focus solely on the risks associated with the use of the product and do not include the manufacturing or supply chain phases [37,39,40]. While GMPs have traditionally focused on ensuring quality standards for medical products, an emerging body of literature and increasing global interest suggest a potential for leveraging this process to address environmental harms [40].

#### 3.2.2. Incorporate GHG Emissions into Requirements for Market Access

To date, market access regulators have expressed no interest in assessing GHG emissions from medical products or the companies producing them. Though not readily connected to the purpose underpinning market access review, agencies could explore the potential to secure standardized GHG emissions reports as part of the market access review process. Relying on companies to voluntarily report GHG emissions information is not sufficient to address the climate crisis; therefore, regulations need to be introduced for the mandatory reporting of emissions from pharmaceuticals. These reporting requirements would require producers to measure, track, and report on emissions at numerous points in the lifecycle to identify “hotspots” and to reduce emissions [41].

#### 3.2.3. Address Environmental Risk Assessment Backlog and Data Availability

Whether or not regulators use environmental data to manage entry into, or use within, the market, there is a currently an untapped opportunity to use the power of regulation to make robust data consistently available. For example, data on the ecotoxicity risks of pharmaceuticals are missing for approximately 85% of the products currently in use [42]. This lack of data is primarily attributed to the fact that the majority of medicines currently used have been in use for more than 20 years (before the introduction of legislation requiring environmental risk assessments (ERA)) [42]. Jurisdictions with this legislation (notably, the US, EU, and Canada) have made little headway in conducting ERAs for legacy products. This lack of data has hindered efforts to identify and assess the ecotoxicological risks of pharmaceuticals at all levels and to conduct comparative analysis to identify environmentally preferable products.

### 3.3. Prescribers

While greener pharmaceutical products, sustainable operations, effective regulation, and rigorous and transparent data must precede environmentally preferable medicine selection, prescribers can do much to reduce environmental harms by reducing inappropriate prescribing and overuse of pharmaceuticals. Prescribers have a critical—yet largely unrealized—opportunity to prioritize appropriate prescribing, which has the power to substantially reduce both upstream and downstream GHG emissions and ecotoxicity pollution from pharmaceuticals.

#### 3.3.1. Reduce Prescribed Medicines

Overprescribing and overuse of medicines generates environmental harms without improving patient or population health [13,35,43,44,45]. An emerging body of literature is reporting on the environmental co-benefits of reducing low-value prescribing, namely, eliminating related ecosystem pollution and GHG emissions from practices that already offer low or no value to patients [46,47]. Prescribers—physicians, nurses, and pharmacists—are the gatekeepers of the majority of pharmaceuticals. Decidedly, the most beneficial action to reduce environmental harm is to reduce the amount of pharmaceuticals prescribed and sold [21]. In addition to reducing inappropriate prescribing, prescribers can leverage strategies such as medication reviews and deprescribing to reduce excess drug use (particularly important for the aging population, where polypharmacy is an issue [11]). Also, treatment strategies, such as initiation packs containing smaller amounts of different dosages can be used to determine the optimal dose for the patient [38,48], or deferred prescriptions, whereby patients are given a prescription for use at a later date if their symptoms do not improve, can reduce pharmaceutical dispensing without negatively impacting clinical outcomes [49,50]. Prescribers engaging with patients regarding the importance of appropriate medicine use is key to implementing these changes [11,17,21].

#### 3.3.2. Prescribe Non-Pharmaceutical Interventions

Healthcare providers also have the ability to “prescribe” non-pharmaceutical interventions, such as social prescribing, including green and blue prescribing (i.e., nature-based prescribing [17]), as alternatives to traditional pharmaceutical treatments, thereby reducing both reliance on medications and the subsequent pharmaceutical pollution. Social prescribing, for instance, offers diverse non-pharmacologic interventions, including engagement with community, arts, and nature, which can contribute to improving patients’ health and well-being [51] and reduce doctor visits and broader health system utilization [52]. Green and blue versions of social prescribing have the added potential to promote pro-environmental behaviors among patients and healthcare providers [17].

#### 3.3.3. Navigate Current Data Limitations

Growing awareness of the environmental harms of pharmaceuticals is encouraging prescribers to make their prescribing practices more sustainable [53]. Of note, several medical associations and societies have recently argued that prescribers should be making decisions informed by the relative carbon footprint of drugs [52], preferring medicines with “the lowest environmental impact” where clinical efficacy is equal [54]. Yet with a few exceptions, as discussed above, the data to support such action are missing, not available, or of questionable adequacy. Without full transparency from producers, directed by clear and consistent requirements from regulators, environmentally preferable medicines cannot be determined and therefore selecting environmentally preferable medicines is not possible. Without the necessary upstream actions, the growing prescriber enthusiasm for reducing environmental harms faces substantial limitations and risks detracting from the longstanding and decisively impactful agenda of reducing overprescribing and overuse.

## 4. Conclusions

The environmental harms from pharmaceuticals, primarily through GHG emissions and ecotoxicity pollution, are significant and occur at many points along the product lifecycle. Upstream actors need to both address these harms (through shifts in manufacturing, business practices, and regulations) and produce and make available validated data on GHG emissions and ecotoxicity from pharmaceuticals. While it is not currently possible to make environmentally preferable distinctions for the vast majority of pharmaceuticals, this lack of data does not prevent prescribers from taking important and impactful actions to address pharmaceutical pollution. Prescribers are critical gatekeepers, and by ensuring appropriate use, they can substantially reduce both upstream and downstream GHG emissions and ecotoxicity pollution from pharmaceuticals.

## Figures and Tables

**Figure 1 pharmacy-12-00173-f001:**
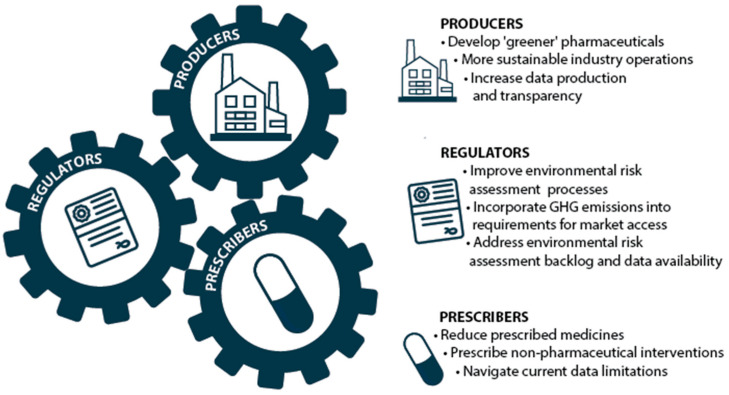
Actions for producers, regulators, and prescribers to address pharmaceutical pollution.

## Data Availability

No new data were created or analyzed in this study. Data sharing is not applicable to this article.

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
