# Peer review of "Tackling Pharmaceutical Pollution Along the Product Lifecycle: Roles and Responsibilities for Producers, Regulators and Prescribers"

_pharmacy, 2024, doi:10.3390/pharmacy12060173_

Round 1
Reviewer 1 Report
Comments and Suggestions for Authors
Journal: Pharmacy
Manuscript ID: pharmacy-3276654
Type: Commentary
Title: Tackling pharmaceutical pollution along the product lifecycle: roles and responsibilities for producers, regulators and prescribers
-------------
The manuscript addresses an important topic - the environmental impact of pharmaceuticals.
However, in my opinion, of the three actors described, prescribers have the least power in reducing GHG emissions and ecotoxicity. Prescribers should be guided first and foremost by the welfare of the patient, rather than choosing a drug with a smaller carbon footprint and less ecotoxicity. This is my opinion. It is also worth quoting numerical examples of some parameters. For example:
L66-L78 - Section 2.1: This section is about the impact on climate change. Therefore, you should give examples of carbon footprint values for some pharmaceuticals (kg CO2eq/kg active substance) to better understand the issue. I suggest adding a small table with examples. Publications with figures are more widely read and more likely to be cited.
Other comments:
Page 1 - headline: change the name and logo of the journal. This is not IJERPH, but Pharmacy.
L71: what anaesthetics? Did you mean nitrous oxide (N2O)? Write it down.
Author Response
The manuscript addresses an important topic - the environmental impact of pharmaceuticals.
However, in my opinion, of the three actors described, prescribers have the least power in reducing GHG emissions and ecotoxicity. Prescribers should be guided first and foremost by the welfare of the patient, rather than choosing a drug with a smaller carbon footprint and less ecotoxicity. This is my opinion. It is also worth quoting numerical examples of some parameters. For example:
L66-L78 - Section 2.1: This section is about the impact on climate change. Therefore, you should give examples of carbon footprint values for some pharmaceuticals (kg CO2eq/kg active substance) to better understand the issue. I suggest adding a small table with examples. Publications with figures are more widely read and more likely to be cited.
Other comments:
Page 1 - headline: change the name and logo of the journal. This is not IJERPH, but Pharmacy.
L71: what anaesthetics? Did you mean nitrous oxide (N2O)? Write it down.
Response:
- We agree with your comment as it aligns with the position presented in this commentary. We argue that prescribers should prioritize appropriate prescribing rather than choosing a drug with a smaller carbon footprint and less ecotoxicity.
- To align with our position stated in the response above and the argument in the commentary regarding the significant lack of data for individual pharmaceuticals we have not presented the issue of GHG emissions at the individual pharmaceutical level.
- Thank you for the suggestion to add a figure. We have added a figure which illustrates the 3 Actors and related recommendations.
- We have transferred the manuscript to the Pharmacy Journal submission template.
- We have added detail regarding anesthetic gases to the manuscript.
Reviewer 2 Report
Comments and Suggestions for Authors
The commentary „Tackling pharmaceutical pollution along the product lifecycle: roles and responsibilities for producers, regulators and prescribers“ systematically demonstrates that addressing environmental considerations along the life cycle of pharmaceuticals requires a coordinated effort among multiple stakeholders. Producers, regulators and prescribers have distinct roles and responsibilities in reducing both greenhouse gas (GHG) emissions and ecotoxicity pollution caused by pharmaceuticals. These roles and responsbilities are clearly stated and explained in this commentary. Hopefully this will lead to deeper analysis and concrete solutions to some of the appointed considerations.
Author Response
The commentary „Tackling pharmaceutical pollution along the product lifecycle: roles and responsibilities for producers, regulators and prescribers“ systematically demonstrates that addressing environmental considerations along the life cycle of pharmaceuticals requires a coordinated effort among multiple stakeholders. Producers, regulators and prescribers have distinct roles and responsibilities in reducing both greenhouse gas (GHG) emissions and ecotoxicity pollution caused by pharmaceuticals. These roles and responsbilities are clearly stated and explained in this commentary. Hopefully this will lead to deeper analysis and concrete solutions to some of the appointed considerations.
Response: Thank you kindly for your thoughtful review of our manuscript.
Reviewer 3 Report
Comments and Suggestions for Authors
The authors prepare a comment on tackling pharmaceutical pollution along the product lifecycle, which is a critical topic in pharmacy as well as in the whole environment. The work addresses the pharmaceutical contributions to GHG emissions and ecotoxicity pollution. The authors discussed the role of producers, regulators and prescribers in the circulation of pharmaceutical compounds. It should be emphasized that social campaigns (in the media) encouraging society to reach for pharmaceutical products play an important role in the abuse of pharmaceuticals.
Author Response
The authors prepare a comment on tackling pharmaceutical pollution along the product lifecycle, which is a critical topic in pharmacy as well as in the whole environment. The work addresses the pharmaceutical contributions to GHG emissions and ecotoxicity pollution. The authors discussed the role of producers, regulators and prescribers in the circulation of pharmaceutical compounds. It should be emphasized that social campaigns (in the media) encouraging society to reach for pharmaceutical products play an important role in the abuse of pharmaceuticals.
Response: Thank you for this suggestion. We have added text to illustrate the prescriber’s role in educating patients about appropriate use of pharmaceuticals.
Reviewer 4 Report
Comments and Suggestions for Authors
First of all I would like to congratulate the authors, they raise a very important issue. Pharmaceutical products place a heavy burden on the environment, both during production and use. We all have an important role to play in implementing change, which, as a pharmaceutical industry, is probably best influenced from the user side, which requires knowledgeable and informed people.
These types of thought-provoking and informative articles are particularly useful.
The work is clear, logical and readable. I have a constant feeling of lack while reading it, but that is probably due to my pessimistic attitude.
The only negative point I can make is that about a fifth of the references are more than 10 years old.
Author Response
First of all I would like to congratulate the authors, they raise a very important issue. Pharmaceutical products place a heavy burden on the environment, both during production and use. We all have an important role to play in implementing change, which, as a pharmaceutical industry, is probably best influenced from the user side, which requires knowledgeable and informed people.
These types of thought-provoking and informative articles are particularly useful.
The work is clear, logical and readable. I have a constant feeling of lack while reading it, but that is probably due to my pessimistic attitude.
The only negative point I can make is that about a fifth of the references are more than 10 years old.
Response: We included seminal (and often older) papers to illustrate the depth of the field, but were also conscious to include the most up-to-date literature, which is reflected in the fact that over 50% of the references are 2022 or later.